# Spatial Shifts of Reflected Light Beam on Hexagonal Boron Nitride/Alpha-Molybdenum Trioxide Structure

**DOI:** 10.3390/ma17071625

**Published:** 2024-04-02

**Authors:** Song Bai, Yubo Li, Xiaoyin Cui, Shufang Fu, Sheng Zhou, Xuanzhang Wang, Qiang Zhang

**Affiliations:** 1Key Laboratory for Photonic and Electronic Bandgap Materials, Ministry of Education, School of Physics and Electronic Engineering, Harbin Normal University, Harbin 150025, China; hsdbs2000@126.com (S.B.); yuboli1998@163.com (Y.L.); 13104688811@163.com (X.C.); xzwang696@126.com (X.W.); 2Department of Basic Courses, Guangzhou Maritime University, Guangzhou 510725, China; zhousheng_wl@126.com

**Keywords:** Goos–Hänchen shift, Imbert–Fedorov shift, α-MoO_3_

## Abstract

This investigation focuses on the Goos–Hänchen (GH) and Imbert–Fedorov (IF) shifts on the surface of the uniaxial hyperbolic material hexagonal boron nitride (hBN) based on the biaxial hyperbolic material alpha-molybdenum (α-MoO_3_) trioxide structure, where the anisotropic axis of hBN is rotated by an angle with respect to the incident plane. The surface with the highest degree of anisotropy among the two crystals is selected in order to analyze and calculate the GH- and IF-shifts of the system, and obtain the complex beam-shift spectra. The addition of α-MoO_3_ substrate significantly amplified the GH shift on the system’s surface, as compared to silica substrate. With the p-polarization light incident, the GH shift can reach 381.76λ_0_ at about 759.82 cm^−1^, with the s-polarization light incident, the GH shift can reach 288.84λ_0_ at about 906.88 cm^−1^, and with the c-polarization light incident, the IF shift can reach 3.76λ_0_ at about 751.94 cm^−1^_._ The adjustment of the IF shift, both positive and negative, as well as its asymmetric nature, can be achieved by manipulating the left and right circular polarization light and torsion angle. The aforementioned intriguing phenomena offer novel insights for the advancement of sensor technology and optical encoder design.

## 1. Introduction

The study of the kinematic effect and physical mechanism of hyperbolic materials has gained increasing popularity in recent years [1,2,3,4]. The momentum space exhibits hyperbolic behavior on the equifrequency plane. The signs of the principal components of the dielectric tensor are significant [5], expressed as diag (*ε_x_*, *ε_y_*, *ε_z_*). When *ε_x_* ≠ *ε_y_* = *ε_z_*, the material is uniaxial hyperbolic material, where the characteristic feature of this material is the presence of hexagonal boron nitride [6,7,8]. The system possesses two distinct Reststrahlen frequency bands (RB), or RB-I corresponding to the longitudinal principal-value of permittivity and RB-II related to the transverse principal values. When *ε_x_* ≠ *ε_y_* ≠ *ε_z_*, the material is biaxial hyperbolic material, where the characteristic feature of this material is the presence of alpha-molybdenum (α-MoO_3_). This crystal exhibits van der Waals bonding and also possesses an asymmetric crystalline structure, with three dielectric tensor principal values corresponding to three Reststrahlen frequency bands [9,10]. In contrast to hBN, the three Reststrahlen frequency bands of alpha-molybdenum trioxide exhibit overlapping regions that are theoretically more anisotropic. Due to their distinctive optical properties, these materials have found extensive applications in various fields such as surface phonon polaritons [11,12,13,14,15,16,17], nanoimaging technology [18,19,20], sensors [21], and waveguides [22,23].

The existing studies indicate that the beam reflected and transmitted from the surface of the medium will exhibit a certain degree of spatial shift. The GH shift occurs in the incident plane, while the IF shift is perpendicular to the incident plane [24]. The GH shift, initially proposed by Newton, refers to a specific displacement between the incident and reflected points of a beam in the incident plane during total reflection. This phenomenon was subsequently experimentally confirmed by Goos and Hänchen in 1947 [25]. The shift primarily arises from the dispersion properties of the reflection coefficient and refraction coefficient. The phenomenon of the shift occurs when the incident beam is fully reflected at the interface, resulting in both a transverse shift within the incident plane and a longitudinal shift perpendicular to the incident plane. The shift primarily arises from the spin-orbit interaction [26]. The two types of shift mentioned above have found wide applications in various fields, including biosensors [27], temperature sensors [28], and high-sensitivity solution concentration measurement [29]. Consequently, they have garnered significant attention from scholars. Deng et al. [30] conducted a study on the beam shift occurring at the surface of a graphene-ITO/TiO_2_/ITO sandwich structure. The GH shift of a Gaussian beam incident on a symmetric structure containing two polar dielectric layers separated by a spacer layer was theoretically calculated by Gupta et al. [31]. The study conducted by Li et al. [32] examined the occurrence of the GH shift and IF shift on the surface of an hBN system that was coated with black phosphorus.

The van der Waals structures offer a platform for the design of quantum materials through the manipulation of weakly bonded atomic layers, employing techniques such as twisting and stacking [33,34]. The utilization of hyperbolic materials is widespread due to their unique structural characteristics, which effectively enhance their in-plane anisotropy through torsional forces. The topological polaron and photon magic angle of the twist bilayer α-MoO_3_ molecular layer were experimentally determined by Hu et al. [35]. The tunable phonon polaron in the twisted molybdenum trioxide system was experimentally determined by Chen et al. [36].

In this paper, we have chosen an α-MoO_3_-based hBN thin film system and successfully achieved enhanced anisotropy within the hBN plane through torsion. The hBN is a natural uniaxial hyperbolic material, while α-MoO_3_ is a natural biaxial hyperbolic material. The Reststrahlen frequency band of these two materials partially overlap. Consequently, the two materials are combined to form a heterostructure. The optical properties at the system’s surface differ significantly from those of a single medium. These interface optical properties can be observed through beam shift on the system’s surface. The shift of the hBN surface beam has been investigated using both the transmission matrix method and the center beam method. The α-MoO_3_ substrate demonstrates enhanced anisotropy and enables greater beam shift compared to the SiO_2_ substrate. Studies on the beam shift using α-MoO_3_ as a substrate have been limited. Therefore, beam shifts on the surface of the hBN/α-MoO_3_ heterostructure is studied. The GH shift on the surface of the hBN/α-MoO_3_ system is enhanced by approximately 15 times compared to graphene/hBN [37], without requiring an increase in bias voltage for the GH shift adjustment.

## 2. Theoretical Model

The hBN/α-MoO_3_ material structure is fabricated, as depicted in Figure 1. The x–z plane has been designated as the incident plane. The x–y plane is selected as the anisotropic plane. The solid blue line represents both the incident light and the reflected light, with the incidence angle set to θ. The hBN film has a thickness of *d*, and its anisotropic axis is torsioned by an angle φ relative to the direction of the incident plane. The three crystallographic directions of the semi-infinite α-MoO_3_ substrate are denoted as x, y, and z, respectively. Herein, the x direction corresponds to the [100] crystallographic direction, the y direction corresponds to the [010] crystallographic direction, and the z direction corresponds to the [001] crystallographic direction. The polarization angle (*β*) of incident light determines its polarization state. When *β =* 0°, the incident light is p-polarization. At *β =* 90°, it becomes s-polarization. For *β =* 45°, the incident light exhibits circular polarization. Δ_X_ and Δ_Y_ denote the GH and IF shifts, respectively. Under this circumstance, the reflected light beam on the material surface will generate a transverse GH shift and longitudinal IF shift in comparison to geometrically reflected light.

The hBN film is twisted at a specific angle with respect to the incident plane. The dielectric tensor matrix is represented by its equivalent form
(1)$$\varepsilon \;=\;\varepsilon_{0}\left(\begin{array}{ccc} \varepsilon_{xx} & \varepsilon_{xy} & 0\\ \varepsilon_{xy} & \varepsilon_{yy} & 0\\ 0 & 0 & \varepsilon_{c} \end{array}\right),$$
where $\varepsilon_{xx}=\varepsilon_{l}\cos^{2}\varphi +\varepsilon_{t}\sin^{2}\varphi$, $\varepsilon_{yy}=\varepsilon_{t}\cos^{2}\varphi +\varepsilon_{l}\sin^{2}\varphi$, and $\varepsilon_{xy}=\left(\varepsilon_{t}-\varepsilon_{l}\right)\cos\varphi \sin\varphi$.

The determination of *k_±_* in hBN and *k_o,e_* in α-MoO_3_ can be achieved by solving Maxwell’s equations and wave equations
(2)$$k_{\pm }^{2}=\frac{1}{2a}(-b\pm \sqrt{b^{2}-4ac}),$$
(3)$$k_{o}=\sqrt{\varepsilon_{y}f^{2}-k_{x}^{2}},$$
(4)$$k_{e}=\sqrt{\left(\varepsilon_{x}\varepsilon_{z}f^{2}-\varepsilon_{x}k_{x}^{2}\right)/\varepsilon_{z}}.$$The detailed solution procedure is in Appendix A.

The electromagnetic boundary condition dictates that the parallel components of both electric and magnetic fields remain continuous across any interface, thereby enabling the determination of the transmission matrix *T*. The detailed solution procedure can be found in Appendix B. We obtain $E_{x,y}^{I,R}=TF_{x,y}=T_{1}T_{2}^{-1}F_{x,y}$. The representation of each element in the transmission matrix *T* is denoted as *t_mn_*. The expressions of the reflection field components are
(5)$$E_{y}^{R}=\frac{t_{11}t_{23}-t_{21}t_{13}}{t_{11}t_{33}-t_{13}t_{31}}E_{x}^{I}+\frac{t_{21}t_{33}-t_{23}t_{31}}{t_{11}t_{33}-t_{13}t_{31}}E_{y}^{I}=a_{11}E_{x}^{I}+a_{22}E_{y}^{I},$$
(6)$$E_{x}^{R}=\frac{t_{11}t_{43}-t_{41}t_{13}}{t_{11}t_{33}-t_{13}t_{31}}E_{x}^{I}+\frac{t_{41}t_{33}-t_{43}t_{31}}{t_{11}t_{33}-t_{13}t_{31}}E_{y}^{I}=a_{21}E_{x}^{I}+a_{22}E_{y}^{I},$$
while the reflective electric field (*z*-component) can be denoted by $E_{z}^{R}=\tan(\beta )E_{x}^{R}$.

The amplitude of the electric field for the transverse electric (TE) wave with s-polarization is accurately represented as $E_{y}^{R}$, while in the reflected beam, the two components of the field amplitude for the transverse magnetic (TM) wave with p-polarization are denoted by $E_{x}^{R}$ and $E_{z}^{R}$. The reflective electric field is represented in matrix form using the O-xyz coordinate system. To accurately address the GH and IF shifts in the reflected beam, it is crucial to apply rotational transformations to both the incident and reflected central waves. According to the geometric correlation, the relationship between the transformation of the incident and reflected electric field can be obtained as $E_{x}^{I}=E_{p}^{I}\cos(\beta )$, $E_{y}^{I}=E_{s}^{I}$, $E_{z}^{I}=-E_{p}^{I}\sin(\beta )$, $E_{p}^{R}=-E_{x}^{R}\cos(\beta )-E_{z}^{R}\sin(\beta )$, $E_{s}^{R}=E_{y}^{R}$, and $E_{z}^{R}=\cot(\beta )E_{x}^{R}$. Consequently, the correlation equation between the incident and reflective fields in the beam coordinate systems can be formulated as
(7)$$\left(\begin{array}{c} E_{p}^{R}\\ E_{s}^{R} \end{array}\right)=\left(\begin{array}{cc} -a_{11} & -\frac{a_{12}}{\cos(\beta )}\\ a_{21}\cos(\beta ) & a_{22} \end{array}\right)\left(\begin{array}{c} E_{p}^{I}\\ E_{s}^{I} \end{array}\right).$$

The subscripts “*s*” and “*p*” denote the s-polarization and p-polarization of the incident and reflected central waves, respectively. The previous results were obtained using the central plane wave. However, if we consider the paraxial wave, the incident beam is confined to a narrow range of plane waves around the central wave in momentum space. The incident beam is assumed to be a Gaussian beam, which deviates from the law of reflection. Equation (7) provides that all elements of the reflection coefficient matrix are dependent on *θ* and *φ*. To do so, we can expand the elements of the coefficient matrix to the first order of $\theta_{p}$ in Equations (5) and (6) to solely consider $\theta_{p}$ while ignoring $\varphi_{s}$. If this dependence arises from the anisotropy of the system, it can be anticipated that both GH and IF shifts may occur for a linearly polarization incident beam. By applying Taylor expansion to Equation (7), we can derive
(8)$$\left(\begin{array}{c} E_{p}^{R}\\ E_{s}^{R} \end{array}\right)=\left(\begin{array}{cc} -\left(a_{11}+a_{11}^{\prime }\beta_{p}\right) & -\frac{a_{12}+a_{12}^{\prime }\beta_{p}}{\cos(\beta )}\\ \left(a_{21}+a_{21}^{\prime }\beta_{p}\right)\cos(\beta ) & a_{22}+a_{22}^{\prime }\beta_{p} \end{array}\right)\left(\begin{array}{c} E_{p}^{I}\\ E_{s}^{I} \end{array}\right).$$

Hence, the explanation of the GH shift provided by
(9)ΔGH=12πk0QImER|∂∂θpER,
with $E^{R}=E_{p}^{R}+E_{s}^{R}$ and $Q={\left|E^{R}\right|}^{2}$.

The formula for the IF shift is derived as follows. The polarization direction of the incident or reflected beam can be directly altered by making a slight adjustment to the orientation of the incident plane. The matrix components in Equation (7), where only *φ_s_* is considered and *θ_p_* is omitted, are also affected. For smaller rotation *φ_s_*, the reflected electric field can be written as
(10)$$\left(\begin{array}{c} E_{p}^{R}\\ E_{s}^{R} \end{array}\right)=\left(\begin{array}{cc} s_{11} & s_{12}\\ s_{21} & s_{22} \end{array}\right)\left(\begin{array}{c} E_{p}^{I}\\ E_{s}^{I} \end{array}\right).$$

Elements in this coefficient matrix are obtained to be
(11)$$s_{11}=-a_{11}+\varphi_{s}\left\{\left[a_{12}/\cos\left(\theta \right)+a_{21}\cos\left(\theta \right)\right]/\tan\left(\theta \right)-a_{11}^{’}\right\},$$
(12)$$s_{12}=-a_{12}/\cos\left(\theta \right)+\varphi_{s}\left[\left(-a_{11}+a_{22}\right)/\tan\left(\theta \right)-a_{12}^{’}/\cos\left(\beta \right)\right],$$
(13)$$s_{21}=a_{21}\cos\left(\theta \right)+\varphi_{s}\left[\left(a_{11}-a_{22}\right)/\tan\left(\theta \right)+a_{21}^{’}\cos\left(\theta \right)\right],$$
(14)$$s_{22}=a_{22}+\varphi_{s}\left\{a_{22}^{’}+\left[a_{12}/\cos\left(\theta \right)+a_{21}\cos\left(\theta \right)\right]/\tan\left(\theta \right)\right\}.$$

The IF shift is computed numerically using Equation (10) and its derivative, which is defined as
(15)ΔIF=−12πk0Qsin(θ)ImER|∂∂φsER.

If the surface layer and substrate are replaced by an isotropic material, the matrix elements of Equation (6) will become $a_{12}=a_{21}=0$, $a_{11}$ = −*f_p_*, and $a_{22}$ = *f_s_* (where *f_p_* and *f_s_* are the reflective coefficients of the *p* and *s* waves). For the high-symmetry configurations ($\varphi =0,90^{o}$), the same results can be obtained $a_{12}=a_{21}=0$, $a_{11}=-f_{p}$, and $a_{22}=f_{s}$. Likewise, the expression of the GH and IF shifts upon the isotropic media can be obtained with the same method. Consequently, Equations (10) and (15) can be simplified to a well-known result, which was cited in the relevant reference [24].

## 3. Results and Discussions

The Lorentz model of hBN and α-MoO_3_ is provided to enhance the analysis of beam shift on the system’s surface, which is respectively expressed as
(16)$$\varepsilon_{i}=\varepsilon_{\infty ,i}\left[1+\left(\omega_{LO,i}^{2}-\omega_{TO,i}^{2}\right)/\left(\omega_{TO,i}^{2}-\omega^{2}-i\omega \Gamma \right)\right],$$
and
(17)$$\varepsilon_{j}=\varepsilon_{\infty ,j}\left[1+\left(\omega_{LO,j}^{2}-\omega_{TO,j}^{2}\right)/\left(\omega_{TO,j}^{2}-\omega^{2}-i\omega \Gamma_{j}\right)\right].$$

The parameters are presented in Table 1 and Table 2.

The Reststrahlen frequency band (RB) is defined as the frequency interval in which the product of the transverse dielectric real part and the longitudinal dielectric real part is less than 0. The material exhibits hyperbolic behavior within the residual frequency range and ellipsoidal behavior outside this range. When the real part of the dielectric in only one direction within the residual frequency band is negative, it manifests as a type I hyperbolic material. Conversely, when the real part of the dielectric in both directions within the residual frequency band is negative, it corresponds to a type II hyperbolic material. The relationship between the dielectric real part of hBN and α-MoO_3_ with frequency is numerically simulated based on the given parameters. The relationship between the real dielectric part of hBN and frequency change is illustrated in Figure 2a. Here, *ε_t_* and *ε_l_* represent the transverse and longitudinal dielectric constants, respectively. It can be observed that the hBN crystal exhibits two Reststrahlen frequency bands within the mid-infrared region. The range of *ω* is 760 cm^−1^ < *ω* < 825 cm^−1^ (green region), indicating the presence of a type I hyperbolic material. Similarly, the range of *ω* is 1361 cm^−1^ < *ω* < 1610 cm^−1^ (pink region), suggesting the existence of a type II hyperbolic material. The relationship between the real part of α-MoO_3_’s dielectric and frequency is illustrated in Figure 2b, while *ε_x_*, *ε_y_*, and *ε_z_* represent the dielectric constants of α-MoO_3_ in three orthogonal directions. The alpha-MoO_3_ exhibits five residual frequency bands in comparison to hBN. These include a Type I hyperbolic material with frequencies ranging from 545 cm^−1^ to 820 cm^−1^ (green region), a Type II hyperbolic material with frequencies ranging from 820 cm^−1^ to 851 cm^−1^ (blue region), another Type I hyperbolic material with frequencies ranging from 851 cm^−1^ to 958 cm^−1^ (yellow region), a Type II hyperbolic material with frequencies ranging from 958 cm^−1^ to 972 cm^−1^ (pink region), and finally, a Type I hyperbolic material with frequencies ranging from 972 cm^−1^ to 1004 cm^−1^ (purple region). The region where the real part of the dielectric function approaches zero is referred to as the epsilon-near-zero (ENZ) region, while the region where the dielectric function tends toward infinity is known as the epsilon-near-pole (ENP) region. This will provide crucial theoretical support for our subsequent analysis of the beam shift.

The physical mechanism of the beam shift is investigated by employing Comsol Multiphysics to simulate the beam shift of α-MoO_3_ covered by hBN near the ENP region, as depicted in Figure 3a. The displacement of the incident light and reflected light centers can be observed, accompanied by a significant enhancement and localization of the electric field intensity. The hybridization of hBN and α-MoO_3_ phonon polaron gives rise to the formation of a surface polaron with enhanced professionalism. The |E| profile of the hBN/α-MoO_3_ surface incident and reflected beams is illustrated in Figure 3b. The solid green line depicts the electric field of the incident light. The solid blue line depicts the electric field of the reflected light in a precise and technical manner. The presence of a disparity in height can be observed at the summit of |E| when x = 48 μm. The highly localized electric field strength of the hBN/α-MoO_3_ surface enables the construction and integration of relatively compact devices.

The special case is chosen when the torsion angle *φ* = 0°, under which circumstance the wave vectors in hBN can be expressed as $k_{+}=\sqrt{\varepsilon_{t}f^{2}-k_{x}^{2}}$ and $k_{-}=\sqrt{\varepsilon_{t}f^{2}-\varepsilon_{t}k_{x}^{2}/\varepsilon_{l}}$. The surface GH shift of hBN/SiO_2_ and hBN/α-MoO_3_ systems is investigated in the current study. The solid lines of varying colors in Figure 4 depict different thicknesses of hBN. The relationship between the GH shift on the hBN/SiO_2_ surface and the frequency change is simulated in Figure 4a. In this case, the GH shift on the system’s surface reaches up to 1.8λ_0_. The frequency-dependent variation of the GH shift on the hBN/α-MoO_3_ surface under the p-wave incident is illustrated in Figure 4b. The surface GH shift of the system exhibits rapid transitions from negative to positive within the frequency range of 759.5 cm^−1^ < *ω* < 760 cm^−1^. The GH shift can achieve a maximum value of 381.76λ_0_ at the frequency *ω* = 759.82 cm^−1^. At this juncture, in conjunction with Figure 2a, it can be observed that the value of Re(*ε_l_*)≈∞ at this frequency is situated within the ENP region of hBN. The combination of Figure 2a,b reveals an overlap in the residual frequency bands between hBN and α-MoO_3_ within this particular region. Compared with Figure 4a,b, the GH shift on the surface of the hBN/α-MoO_3_ system is enhanced by about 212 times. The surface of the hBN/SiO_2_ system under s-wave irradiation is depicted in Figure 4c, illustrating the observed GH shift. The GH shift value generated on the system’s surface is negligible. However, due to the pronounced anisotropy of the α-MoO_3_ substrate, the surface of the system also exhibits a significant GH shift under s-wave irradiation, as illustrated in Figure 4d. Under s-wave irradiation, the GH shift on the surface of the hBN/α-MoO_3_ system exhibits a transition from negative to positive within the frequency range of 906.5 cm^−1^ < *ω* < 907.25 cm^−1^, with a peak value of 288.84λ_0_ observed at *ω* = 906.88 cm^−1^. This frequency falls within the Reststrahlen frequency band of α-MoO_3_.

In the analysis of the beam shift, it is crucial to consider not only the distinct residual frequency bands of the material but also place significant emphasis on the reflectivity parameter. The reflection coefficient of p-polarization light is as follows: *R_p_ =* (*εfcosβ* − *k_z_*)/(*εfcosβ* − *k_z_*). The reflection coefficient of s-polarization light is being studied: *R_s_ =* (*fcosβ* − *k_z_*)/(*fcosβ* + *k_z_*). The critical angle is defined as *sin*^2^*β < ε*, where ε is a positive value between 0 and 1. The Brewster angle can be defined as the angle of incidence at which the reflectance R ≈ 0 corresponds to a specific frequency. The reflectance of the p-wave and s-wave exhibits frequency-dependent variations for an hBN thickness of 43 nm, as illustrated in Figure 4. Analysis of Figure 5a,b reveals that the reflectance R ≈ 0 coincides with the occurrence of the GH shift peak at frequencies *ω* = 759.82 cm^−1^ and *ω* = 906.88 cm^−1^, respectively, for incident p-polarization and s-polarization light, both aligning with the Brewster angle.

The previous discussion focused on the GH shift of p-polarization and s-polarization light, followed by a numerical simulation of the IF shift of the system’s surface. The IF shift of the system surface is primarily attributed to the interaction of spin orbital angular momentum, which arises from the interaction between elliptically polarization beams comprising p-polarization light and s-polarization light. Consequently, our focus lies in discussing the IF shift under circularly polarization light incidence. When the polarization angle *β* = 45°, it corresponds to right-handed circularly polarization light. Conversely, when *β* = −45°, it corresponds to left-handed circularly polarization light. The frequency variation of the IF shift for incident right-handed circular-polarization light and left-handed circular-polarization light on hBN surfaces with varying thicknesses is investigated under two specific torsion angle conditions, *φ* = 0° and *φ* = 90°, as illustrated in Figure 6. The dielectric tensor of hBN is represented as diag(*ε_l_*, *ε_t_*, *ε_t_*) and diag(*ε_t_*, *ε_l_*, *ε_t_*) at two torsion angles, respectively. Similarly, the dielectric tensor of α-MoO_3_ is expressed as diag(*ε_x_*, *ε_y_*, *ε_z_*), both of which are denoted by diagonal matrices. Under identical conditions, it can be observed that the positive and negative symmetry of the IF shift induced by right-handed circularly polarization light and left-handed circularly polarization light is attributed to the material’s dielectric tensor in the form of a diagonal matrix.

We conducted numerical simulations to investigate the reflectance of the system under circularly polarization light incidence, as depicted in Figure 7. Our findings indicate that both right-handed and left-handed circularly polarization light have no impact on the reflectance of the system under identical conditions. Moreover, due to a small incident angle θ, when the torsion angle *φ* is 0° and the hBN thickness is 100 nm at a frequency *ω* = 852 cm^−1^, the reflectance R approaches approximately 0.5. In this scenario, the induced IF shift can reach up to 1.55λ_0_. Conversely, when *φ* = 90° with an hBN thickness of 500 nm and frequency *ω* = 852 cm^−1^, we observe a similar reflectance R ≈ 0.5 leading to an IF shift of approximately 2.53λ_0_. Notably, the high reflectivity of the material eliminates the need for weak measurement technology during observation, highlighting its practical application value.

As mentioned above, the positive and negative symmetry of the IF shift generated by right-handed and left-handed circular-polarization light under identical conditions is attributed to the diagonal form of the dielectric tensor matrix of the material. The expression form of the dielectric matrix can be altered through twisting. Subsequently, hBN with thicknesses of 100 nm and 500 nm, respectively, were selected. By varying the torsion angle, we explored the positive and negative symmetry of the IF shift as depicted in Figure 8. It is observed that altering the torsion angle of hBN’s anisotropic axis resulted in a nondiagonal dielectric tensor matrix for hBN, and therefore eliminated the positive and negative symmetry in the IF shift on system surfaces under identical conditions. This unique optical phenomenon provides novel insights for designing and fabricating optical encoders. The surface anisotropy of the system is enhanced due to the specific twisting angle of hBN’s main optical axis. The IF shift is observed in the reststrahlen frequency band of α-MoO_3_, as depicted in Figure 8b,d. The IF shift with frequency and twisting angle is depicted in Figure 8b when right-handed circular polarizing light is incident. Narrow peaks are observed within the Reststrahlen frequency band of α-MoO_3_, specifically between frequencies 700 cm^−1^ < *ω* < 750 cm^−1^ and 850 cm^−1^ < *ω* < 900 cm^−1^. The IF shift with frequency and twisting angle is depicted in Figure 8d when left-handed circular polarizing light is incident on the surface of the system. Narrow peaks were observed 800 cm^−1^ < *ω* < 850 cm^−1^. The frequency range in question is situated at the interface between the Reststrahlen frequency band of hBN and α-MoO_3_, specifically within the ENZ region of α-MoO_3_. It should be noted that α-MoO_3_ exhibits Type II hyperbolic material behavior with a notable degree of anisotropy.

Next, we investigate the maximum value of the system’s IF shift, as depicted in Figure 9a,b. It is observed that for an hBN thickness of 100 nm, torsion angle *φ* = 60°, incident angle *θ* = 10°, and right-circularly polarization incident light, the system exhibits an IF shift of 3.76λ_0_ at frequency *ω* = 751.94 cm^−1^, thereby enhancing the in-plane hyperbolism of hBN. Additionally, when the hBN thickness is increased to 500 nm with a torsion angle *φ* = 45° and left-handed circularly polarization incident light, the IF shift can reach 3.18λ_0_ at frequency *ω* = 820 cm^−1^ within the Reststrahlen frequency band of Type II hyperbolic material α-MoO_3_. Based on Figure 2a,b analysis reveals that this frequency corresponds to an overlap between the Reststrahlen frequency band of hBN and α-MoO_3_.

## 4. Conclusions

This present study investigates the GH shift and IF shift on the hBN/α-MoO_3_ by using the transmission matrix method and the central beam method. The formula for beam shift on the structure’s surface is derived using a torsion coordinate system, followed by numerical simulation. Under p-polarization light incidence, GH shift reaches its maximum value of 381.76λ_0_ at frequency *ω* = 759.82 cm^−1^, with a corresponding reflectance R ≈ 0 and an incidence angle equal to the Brewster angle. We also examine the IF shift of right-handed and left-handed circularly polarization light reflected on the hBN/α-MoO_3_ system’s surface. When the dielectric tensor of the system is diagonal, both right-handed and left-handed circularly polarization light exhibits positive and negative symmetry under identical conditions with matching frequencies. However, this symmetry ceases to exist when torsion coordinates alter the dielectric tensor to a nondiagonal matrix. By applying torsion at *d* = 100 nm and *φ* = 60°, an IF shift of 3.76λ_0_ can be achieved. These findings offer theoretical guidance for advancing novel nano-optical devices and optical encoders.

## Figures and Tables

**Figure 1 materials-17-01625-f001:**
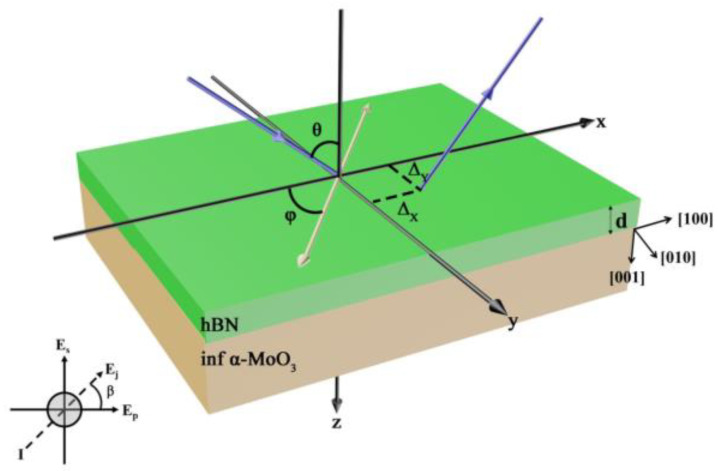
The configuration diagram illustrates the structure of a semi-infinite alpha-molybdenum trioxide substrate with an hBN film coating.

**Figure 2 materials-17-01625-f002:**
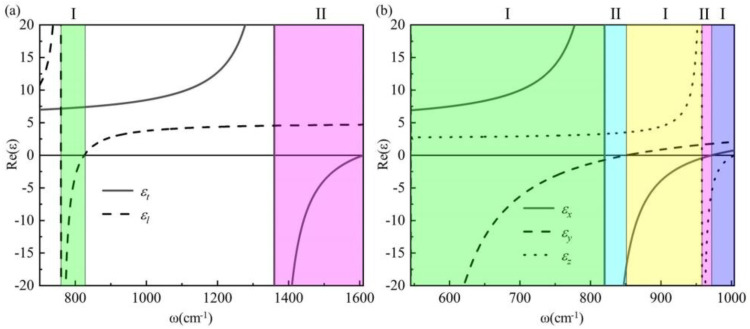
Illustrates the relationship between the real part of the dielectric constant and frequency for: (**a**) hBN crystal; (**b**) α-MoO_3_ crystal.

**Figure 3 materials-17-01625-f003:**
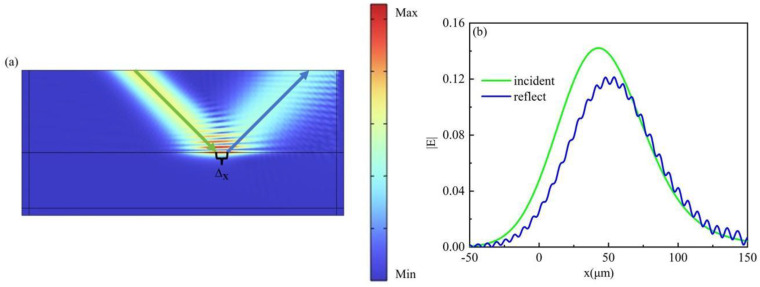
(**a**) The simulation of the GH shift on the hBN/MoO3 surface. (**b**) The distribution of electric field for incident and reflected light on the surface of the ENP region.

**Figure 4 materials-17-01625-f004:**
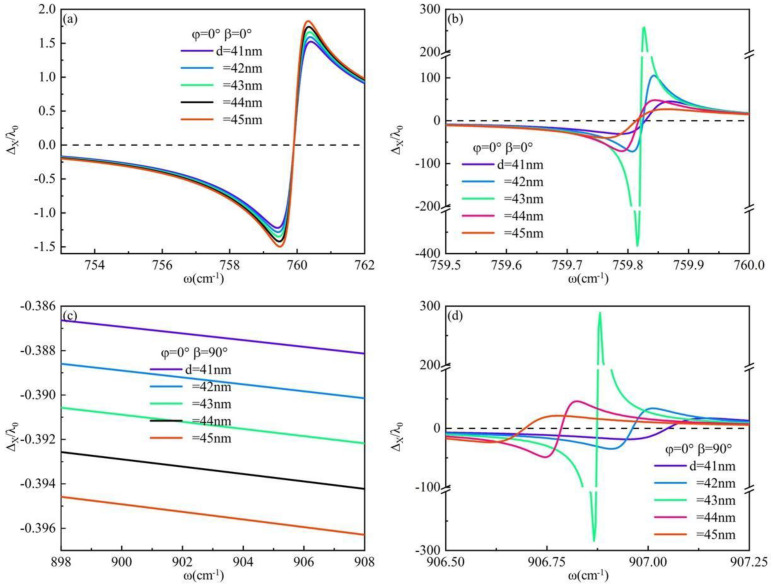
The relationship between the GH shift on the system surface and frequency when *φ* = 0°: (**a**) p-polarization light incident on hBN/SiO_2_; (**b**) p-polarization light incident on hBN/α-MoO_3_; (**c**) s-polarization light incident on hBN/SiO_2_; (**d**) s-polarization light incident on hBN/α-MoO_3_.

**Figure 5 materials-17-01625-f005:**
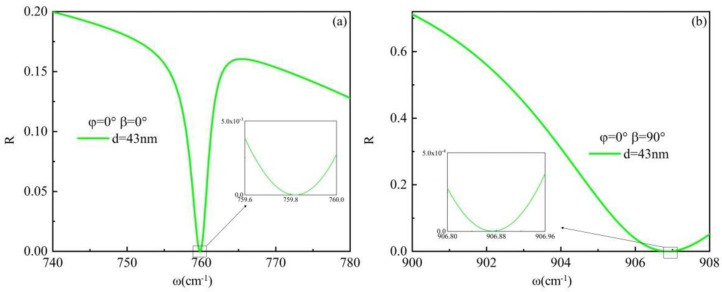
When *φ* = 0° and *d* = 43 nm, the reflectance of the system surface varies with frequency under: (**a**) p-polarization light incidence; (**b**) s-polarization light incidence.

**Figure 6 materials-17-01625-f006:**
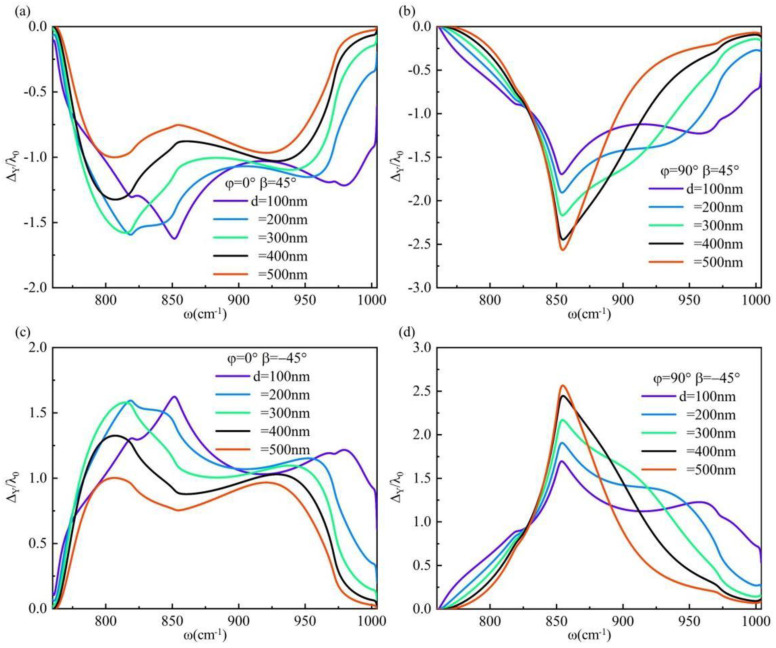
The frequency-dependent variation of the shift of an hBN surface with different thickness is observed at incidence angles *θ* = 10°: (**a**) *φ* = 0°, *β* = 45°; (**b**) *φ* = 90°, *β* = 45°; (**c**) *φ* = 0°, *β* = −45°; and (**d**) *φ* = 90°, *β* = −45°.

**Figure 7 materials-17-01625-f007:**
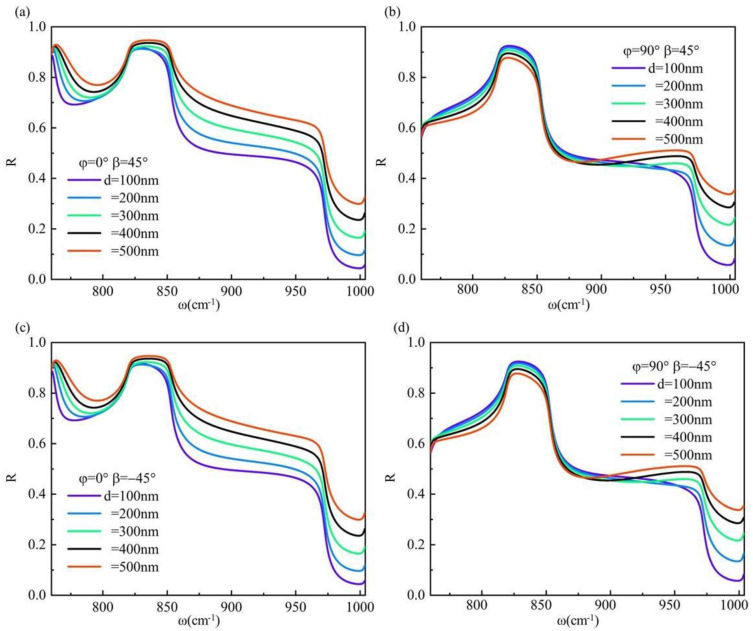
When the hBN thickness of the system varies, the surface reflectance R exhibits frequency-dependent changes under an incidence angle *θ* = 10°: (**a**) *φ* = 0°, *β* = 45°; (**b**) *φ* = 90°, *β* = 45°; (**c**) *φ* = 0°, *β* = −45°; (**d**) *φ* = 90°, *β* = −45°.

**Figure 8 materials-17-01625-f008:**
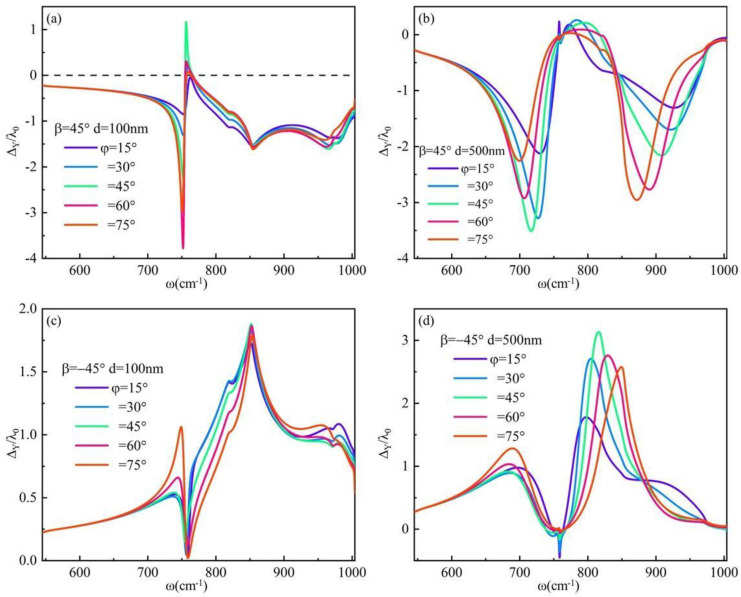
The IF shift of the system surface exhibits frequency-dependent variations when the torsion angle is altered, with an incident angle *θ* = 10°: (**a**) *β* = 45°, *d* = 100 nm; (**b**) *β* = 45°, *d* = 500 nm; (**c**) *β* = −45°, *d* = 100 nm; (**d**) *β* = −45°, *d* = 500 nm.

**Figure 9 materials-17-01625-f009:**
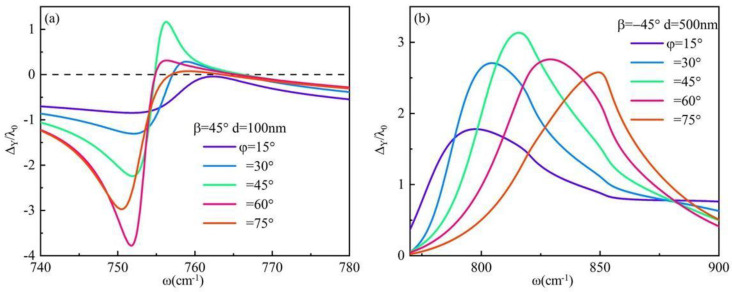
When the torsion angle is adjusted, the frequency-dependent shift of the interface on the system’s surface varies. The incident angle *θ* is set at 10°: (**a**) *β* = 45°, *d* = 100 nm; (**b**) *β* = −45°, *d* = 500 nm.

**Table 1 materials-17-01625-t001:** The parameters of hBN [38].

	*t*	*l*
*ε_∞,i_*	4.52	4.95
*ω_TO,i_* (cm^−1^)	1610	825
*ω_LO,i_* (cm^−1^)	1360	760
Γ (cm^−1^)	2	

**Table 2 materials-17-01625-t002:** The parameters of α-MoO_3_ [39,40].

	*x*	*y*	*z*
*ε_∞,j_*	4.0	5.2	2.4
*ω_TO,j_* (cm^−1^)	972	851	1004
*ω_TO,j_* (cm^−1^)	820	545	958
Γ*_j_* (cm^−1^)	4	4	2

## Data Availability

The references in this paper have been explained.

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
