# Peer review of "Spatial Shifts of Reflected Light Beam on Hexagonal Boron Nitride/Alpha-Molybdenum Trioxide Structure"

_materials, 2024, doi:10.3390/ma17071625_

Round 1

Reviewer 1 Report

Comments and Suggestions for Authors

It is a scientific mathematical work about the spatial shifts of reflected light beam on hexagonal boron-nitride/alpha-molybdenum trioxide structure. I recommend the publication after a minor revision.

I suggest reducing the Theoretical model section. In a research manuscript, there are same issues such discuss the expression of the solution of a quadratic equation that are not necessary. One alternative is to move part of the theoretical model description to an annex.

In relation with the previous comment, the authors can improve discussion about the boundary conditions needed to determine the transmission matrix for z = 0 and z = d.

Figure 8: There are some narrow peaks (8b, 8d). Why? The authors should improve discussion.

Finally, I think it is necessary to remark the novelty by comparing with recent scientific literature.

Comments on the Quality of English Language

Minor editing of English language required

Author Response

Dear Reviewer

On behalf of my co-authors, we thank you very much for offering us an opportunity to revise our manuscript (ID: materials-2775485). We appreciate the editor and reviewers very much for their positive and constructive comments and suggestions on our manuscript. Those comments are valuable and very helpful for revising and improving our manuscript. We have finished our revised version. The changes and responses to the comments are given at PDF called "response letter" .

Yours sincerely!

Qiang Zhang

Reviewer 2 Report

Comments and Suggestions for Authors

The authors have reported on the spatial shifts of reflected light beam on hexagonal boron nitride/alpha-molybdenum trioxide structure.

I have the following comments on the manuscript.

1.     First of all, the submission type of manuscript is Review. However, in the main text it is mentioned as paper.

As the authors discussed the theoretical analysis of the hBN and α-MoO3 system, it should consider as original article. This is a mistake in manuscript submission, which should be corrected accordingly.

Authors have written in the introduction part as follows,

In this paper, We have chosen an α-MoO3 based hBN thin film system and successfully achieved enhanced anisotropy within the hBN plane through torsion. The shift of the hBN surface beam has been investigated using both the transmission matrix method and the center beam method. The α-MoO3 substrate demonstrates enhanced anisotropy and enables greater beam shift compared to the SiO2 substrate”

 2.     Bulk α-MoO3 forms an orthorhombic crystal, whereas the hBN are layered crystalline materials, how does the interface influence on the reflected light beam?

Is there any other compatible material system for such applications, rather than the α-MoO3 heterostructure with hBN?

Regarding this aspect, authors can provide brief discussion in the introduction part.

 3.     English typos should be corrected.

“we” should small letter, in the sentence In this paper, We have…”

 I think the theoretical analysis of the hBN and α-MoO3 system is interesting prospect for application in optoelectrical devices.

I recommend acceptance of the manuscript for publication with major revisions.

Comments on the Quality of English Language

Minor editing of English language required

Author Response

Dear Reviewers:

On behalf of my co-authors, we thank you very much for offering us an opportunity to revise our manuscript (ID: materials-2775485). We appreciate the editor and reviewers very much for their positive and constructive comments and suggestions on our manuscript. Those comments are valuable and very helpful for revising and improving our manuscript. We have finished our revised version. The changes and responses to the comments are given at PDF called "response letter".

Yours sincerely!

Qiang Zhang

Reviewer 3 Report

Comments and Suggestions for Authors

The manuscript reports on the measurement of the Goos-Hänchen  and Imbert-Fedorov shifts on hBN-alpha-MoO3 planar structure. The presented results represent original contribution to the investigation of both GH and IF effects. The manuscript should be published in Materials. I did not notice principal deficiency in the research but I am a bit confused from the real research, which was done. Reading the initial texts and inspecting Figure 1, I expected that the theoretical calculations will be accompanied by appropriate experiments. However, further text changed my meaning and I deduce that all results are based on the numerical simulations of the structure in Fig.1. Authors should clearly inform on the type of works. They should also suggest appropriate experiments, which could be performed to verify predicted results. It is known that the it is difficult to grow heterostructures without significant damage at the interface. Did authors consider such problem? It could significantly degrade their achievements. The material parameters of the compound structure used at the calculations including references should be given.

Comments on the Quality of English Language

English is fine with me.

Typing and formatting drawbacks:

Square brackets for references should not be terminated by a comma.

6th line below eq. (6): 'Equation (9) provide ...'

Author Response

(The authors gave the same response as above.)

Round 2

Reviewer 2 Report

Comments and Suggestions for Authors

 The authors have revised the manuscript as suggested. I would like to recommend acceptance of the manuscript.